# Cancer Stem Cells and Neovascularization

**DOI:** 10.3390/cells10051070

**Published:** 2021-04-30

**Authors:** Fengkai Li, Jiahui Xu, Suling Liu

**Affiliations:** 1Fudan University Shanghai Cancer Center & Institutes of Biomedical Sciences, Cancer Institutes, Fudan University, Shanghai 200032, China; lifengkai@fudan.edu.cn (F.L.); xujiah@mail.ustc.edu.cn (J.X.); 2Key Laboratory of Breast Cancer in Shanghai, The Shanghai Key Laboratory of Medical Epigenetics, Fudan University, Shanghai 200032, China; 3The International Co-Laboratory of Medical Epigenetics and Metabolism, Ministry of Science and Technology, Shanghai Medical College, Fudan University, Shanghai 200032, China

**Keywords:** cancer stem cells, angiogenesis, vascular mimicry, therapeutic strategies

## Abstract

Cancer stem cells (CSCs) refer to a subpopulation of cancer cells responsible for tumorigenesis, metastasis, and drug resistance. Increasing evidence suggests that CSC-associated tumor neovascularization partially contributes to the failure of cancer treatment. In this review, we discuss the roles of CSCs on tumor-associated angiogenesis via trans-differentiation or forming the capillary-like vasculogenic mimicry, as well as the roles of CSCs on facilitating endothelial cell-involved angiogenesis to support tumor progression and metastasis. Furthermore, we discuss the underlying regulation mechanisms, including the intrinsic signals of CSCs and the extrinsic signals such as cytokines from the tumor microenvironment. Further research is required to identify and verify some novel targets to develop efficient therapeutic approaches for more efficient cancer treatment through interfering CSC-mediated neovascularization.

## 1. Introduction

Cancer is a serious disease that threatens human health all over the world. Cancer treatment options include chemotherapy, radiotherapy, surgery, and some newly developed Immunotherapies. However, the efficiency of these treatments strongly depends on multiple factors such as cancer staging, drug-resistance, and even the tumor-microenvironment. Blood vessels play a crucial role in tumor progression to supply the oxygen and nutrients [1]. Until Folkman’s studies provided the evidence that tumors depend on angiogenesis [2], tumors were thought to acquire their blood supply from preexisting blood vessels. However, unfortunately, drugs or compounds aimed to suppress the angiogenesis, such as Bevacizumab blocking vascular endothelial growth factor (VEGF) and VEGF inhibitor vandetanib, have not achieved the desired effects [3,4]. In recent years, a growing number of reports have described that cancer stem cells (CSCs) are involved in drug-resistance and relapse [5,6,7]. CSCs refer to a subpopulation of tumor cells that have abilities to self-renew, differentiate, and seed new tumors [8]. In this regard, CSCs exhibit self-renewal and multilineage differentiation abilities; they might be taking part in tumor-associated angiogenesis via trans-differentiation or forming the capillary-like vascular mimicry (VM) in the tumor microenvironment.

In this review, we will discuss the current knowledges of CSCs participating in VM and angiogenesis in order to understand the underlying mechanisms that lead to the development of more effective therapies for cancer treatment.

## 2. Vascular Mimicry

### 2.1. Molecular Determinants Regulating the Formation of Vascular Mimicry

Regarding the significance of VM in malignant tumor progression, researchers have made great efforts to explore the molecular mechanisms driving the aggressive tumor cells to display the endothelial-like phenotype. Several key molecules have been found to regulate the formation of VM. Seftor et al. have already demonstrated several important cellular and molecular determinants of aggressive melanoma VM, including VE-Cadherin and EphA2 [9]. VE-Cadherin is an adhesion molecule previously thought to be exclusively expressed by endothelial cells [10,11]. However, it was found to be exclusively highly expressed in the aggressive melanoma cells and undetectable in the non-aggressive melanoma cells, supporting a vasculogenic-like patterned network of aggressive melanoma cells in three-dimensional culture. No networks could form when VE-Cadherin was down-regulated [9]. EphA2 is a protein belonging to ephrin receptor subfamily of the protein-tyrosine kinase family. It was also found to be expressed exclusively in the aggressive melanoma cells. Down-regulation of EphA2 abrogated the ability of tumor cells to form patterned networks [9,12] (Hess et al., 2001; Seftor et al., 2002). Furthermore, VE-Cadherin and EphA2 act in a coordinated manner in the regulation of vascular signaling pathway of melanoma cells. VE-Cadherin could interact with EphA2 and, thus, mediate the plasma membrane localization and phosphorylation of EphA2 [13].

Other molecular targets were also reported to participate in VM formation including the matrix matalloproteinases (MMPs) and the focal adhesion kinase (FAK). The microarray gene chip analysis revealed a significant increase of ECM component in aggressive melanoma cells, including the MMP-1, MMP-2, MMP-9, MT1-MMP, and Laminin5. The Laminin5γ2 chain fragmentation by active MT1/MMP and MMP2 is an essential step for aggressive melanoma cells to engage in VM formation [14]. FAK signaling blockade also inhibited the VM formation. It is reported that EphA2 drives the cleavage of Laminin 5 via activation of FAK signaling, which subsequently leads to the activation of extracellular signal-regulated kinase 1 and 2 (ERK1/2) and finally induces the active MT1/MMP and MMP2 through the PI3K pathway [13,15] (Figure 1).

### 2.2. The Role of CSCs in Vascular Mimicry Formation

Besides the molecular determinants mentioned above, it is reported that the tumor cell VM is associated with stem cell characteristics. As early as the elaboration of VM by Maniotis et al., they compared the highly invasive versus poorly invasive melanoma tumor cells through cDNA microarray analysis, and found that the vascular mimetic invasive melanoma cells displayed a pluripotent embryonic-like genotype [16]. Furthermore, Seftor et al. found that vasculogenic mimetic melanoma cells expressed multiple genes that regulated cell plasticity in embryogenesis [9]. In glioblastoma, Ricci-Vitiani et al. found that a significant portion of the vascular endothelium shared the same genomic alteration as tumor cells in tumor xenografts produced from orthotopic or subcutaneous injection of glioblastoma stem-like cells (GSCs) in immunocompromised mice. The vascular endothelium contained a subset of tumorigenic cells that could generate highly vascularized tumors with VM. In addition, in vitro culture of GSCs in the endothelial conditions endowed them with an endothelial-like phenotype [17]. Moreover, the tumor cells lining VM channels were found to express the stem cell factor SOX2 and OCT4 in hepatocellular carcinoma [18], suggesting a new mechanism for CSC-mediated tumor VM formation. Recently, various studies have described the participation of CSCs in the formation of VM in multiple types of cancer. The presence of several CSC markers is associated with VM formation.

#### 2.2.1. CD44^+^ CSCs

The cell surface protein CD44, a transmembrane receptor for hyaluronic acid, has been implicated in a diverse array of cellular function, including cell–cell and cell–matrix interaction, differentiation, proliferation, apoptosis, and motility [19,20]. In breast cancer, CD44^+^CD24^−^ tumor cells were identified as tumorigenic cells. As few as 100 cells of this tumorigenic subpopulation were able to generate tumor in mouse xenograft model [21]. In microvascular endothelial cells, CD44 was demonstrated to play a key role in regulating the cell proliferation and survival via activating downstream Hippo signaling pathway and modulating the expression of CD31 and VE-Cadherin [22]. In addition, CD44 variant expressed on metastatic melanoma cells induced the VE-Cadherin phosphorylation at Y658 and Y731, and, thus, disrupted the endothelial junction assembly, finally promoted the melanoma transendothelial migration, documenting a critical role for CD44 in mediating vascular functions [23]. CD44 is significantly correlated with the VM presence in multiple solid tumors [24,25,26]. Recently, CD44 is also found to significantly correlate with VE-Cadherin expression and VM presence in oral squamous cell carcinoma [26]. In Ewing sarcoma tumors and breast carcinoma tumors, the CD44/c-Met signaling pathway was identified as the key regulator for VM through microarray analysis of aggressive cells (VM forming) and non-aggressive cells (non-VM forming). Both CD44 standard isoform and its splice variant CD44v6 were relevant to VM. Overexpression of CD44 in Ewing sarcoma tumor cells increased its interaction with extracellular matrix ligand hyaluronic acid and facilitated the formation of vasculogenic networks in vitro, whereas CD44 knockdown suppressed it [25], illuminating a critical role of CD44 in the process of VM formation.

#### 2.2.2. ALDH^+^ CSCs

The enzyme aldehyde dehydrogenase 1 (ALDH1) has been shown to be a marker for both normal and malignant breast stem cells. Breast tumor cells with increased ALDH1 expression displayed the stem/progenitor properties and showed the strongest tumorigenic capacity in the mouse xenograft model [27]. Since then, ALDH has been proved to be a CSC marker in several solid tumors [28]. Both the ALDH1 expression and VM presence predicted poor disease-free survival and overall survival of breast cancer patients and colorectal cancer patients. The expression of ALDH1 was strongly associated with the presence of VM in breast cancer, especially in aggressive triple-negative breast cancer [29,30]. A recent study using the triple-negative breast cancer cell line HCC1937/p53 has further shown that the ALDH^+^ cells sorted by fluorescence-activated cell sorting markedly formed VM on Matrigel culture, whereas the ALDH^−^ cells did not [31].

#### 2.2.3. CD133^+^ CSCs

CD133, a pentaspan membrane glycoprotein, is one of the most well characterized markers of the CSCs in multiple types of cancer. Accumulating evidence has shown the importance of CD133 in regulating CSC-mediated tumorigenesis, metastasis, and chemoresistance [32]. CD133 is significantly correlated with VM presence in human renal cell carcinoma [24]. In acute leukemia, the adherent bone marrow stromal cells derived from CD133^+^/CD34^+^ stem cells were able to form the capillary-like structures on Matrigel through secreting the insulin growth factor-1 (IGF-1) [33]. CD133 was also demonstrated to be associated with VM phenotype in a study including 134 samples of breast cancer patients. The CD133^+^ subpopulation of MDA-MB-231 cell line expressed higher level of VM-related genes VE-Cadherin, MMP-2 and MMP-9, and was able to form the vasculogenic-like channels in the matrix culture [34]. The breast tumor cells lining VM channels on Matrigel were also found to express CD133. Via using a three-dimensional reconstructed image to show the spatial relationship between CSCs and VM, Sun et al. has provided direct evidence indicating that tumor cells lining VM channels were derived from CSCs [35].

Additionally, CSCs may provide more VM-related factors to synergize VM formation. For instance, the Nodal protein, a member of the transforming growth factor-β (TGF-β) superfamily, has been reported to be expressed by CSCs to maintain its stem cell-like properties, and promote the VM formation via Smad2/3 signaling pathway in the in vitro study [36,37]. In conclusion, CSCs facilitate the VM formation by promoting the VM-related gene expression and an endothelial-like phenotype, and probably lining VM channels directly. However, the CSC subpopulation is known to be heterogenous. Although several markers have been identified to isolate the CSC subgroups, they are not always completely overlapping. For example, Liu et al. has found that breast CSCs existed in at least two distinct states. Breast CSCs located in the invasive edge of tumors are characterized as CD44^+^CD24^-^ and primarily quiescent, whereas breast CSCs located in the central region are more likely to express ALDH and more proliferative [38]. The heterogeneity of CSCs existed within tumors raises the questions that, which group of CSCs is most responsible for the VM formation, and what are the exact molecular mechanisms of regulating CSC-mediated VM formation. Moreover, the plastic transformation between different states of CSCs makes it much more complex. There is still a long way to go before we thoroughly understand the VM phenomenon.

### 2.3. Therapeutically Targeting Vascular Mimicry

As mentioned above, VM formation is an important aspect of the tumor neovascularization distinguished from the endothelial cell-dependent angiogenesis, playing crucial roles in tumor growth and metastasis. There have been several studies linking angiogenesis-promoting factors to VM. For example, VEGFA could induce the formation of VM in vitro via upregulating the expression of EphA2 and MMPs in ovarian cancer cells [39]. However, the upregulated expression of VEGF was not significant in VM-forming tumor cells compared with other VM-related genes [40]. What is more, the anti-angiogenic agents could not specifically target the vasculogenic CSCs population. Therefore, the therapeutic effects of clinical anti-angiogenic agents on VM are very limited. Both the anti-VEGFA monoclonal antibody Bevacizumab and the endostatin have been reported to have no effect on VM [41,42,43]. The application of Bevacizumab failed to block the vascular structures derived from CD133^+^ CSCs in glioblastoma [43]. In fact, Bevacizumab treatment even led to increased VM in tumors [44]. Accordingly, therapeutic inhibition of VM in combination with other anti-angiogenic therapies can inhibit the tumor vascularization to the utmost extent. Actually, there have been several drugs or compounds that show the VM inhibition effect. For example, application of the cytotoxic drug vincristine in combination with the tyrosine kinase inhibitor dasatinib could delete the VM channels via inhibition of VM-related indicators VE-Cadherin, FAK, PI3K, and MMPs [45]. The natural extract compounds such as Hinokitiol and brucine, also have shown an inhibition effect on VM channels [46,47]. However, currently there are still no specific therapies to inhibit VM formation. Since VM is closely correlated with and regulated by the CSC status, it is possible that therapies targeting the presence of CSCs in combination with the anti-angiogenic therapies may be the best strategy to eliminate the tumor vessel formation.

CSCs enhance the VM formation via IGF-1, VE-Cadherin, MMPs, and FAK pathways.

## 3. CSC-Derived Endothelial Cells

In general, VEGF plays a critical role in the formation of the embryonic circulatory system and growth of blood vessels from pre-existing vasculature [48]. The vasculogenesis relies on VEGF in normal tissues. Especially, in glioblastomas tissue, the glioma stem cells (GSCs) have been reported to promote angiogenesis via releasing high level of VEGF [49,50]. CSCs maintain the potential of transdifferentiating into various cell types. In recent decades, CSCs/tumor cell derived endothelial cells have been reported in a variety of solid tumors [17,51,52,53].

### 3.1. CSCs Transdifferentiate into Endothelial Cells

#### 3.1.1. Glioblastoma

Glioblastoma is one of the most vascular-rich tumors. Ricci-Vitiani et al. reported that the CD31^+^/CD144^+^ endothelial cells from freshly dissociated glioblastoma specimens shared the same chromosomal alterations. Of endothelial cells, 20–90% contain the same chromosomal alterations as tumor cells in glioblastoma. Moreover, the authors observed that human-specific CD31^+^ protein were only expressed in the tissue that formed in subcutaneous xenografts generated by the injection freshly isolated CD133^+^/CD31^−^ glioblastoma cells [17,54]. Similarly, the authors in another group employed GFP labeled glioblastoma mice model and successfully obtained GFP-positive tumor cells with CD31^+^ or CD34^+^ endothelial cell characteristics, particularly in the deep area of the lesions [53]. By mean of the flow cytometry, it is shown that 10–25% of endothelial cells (CD45^−^CD31^+^CD34^+^) were positive for GFP.

The phenomenon observed in those studies indicates that glioma stem cells or glioblastoma cells have the potential of differentiating into endothelial lineages.

#### 3.1.2. Renal Carcinoma

Renal carcinoma is one kind of human malignancies, with high metastatic and poor prognosis. In the cause of study a new drug for renal carcinoma, the authors search for the presence of a tumor-initiating stem cell population in renal carcinomas [52], they separated the renal tumor cells and used the mesenchymal stem cell marker CD105 from human renal carcinomas. Afterwards, they found that CD105^+^ cells but not CD105^−^ cells have the proficiency of differentiation into vWF/KDR/VEGFR3/CD31 positive endothelial cells in vitro; moreover, they observed the direct contribution to the presence of endothelial cells in tumor vessels at SCID mice xenograft. This is the proof that CD105^+^ renal CSCs can not only generate endothelial cells in vitro, but also give rise to vessels with a human origin in vivo.

#### 3.1.3. Breast Cancer

It is well established that CD44^+^CD24^−^/low cells were recognized as breast cancer stem cells [21]. Bussolati et al. employed mammosphere culture to enrich the CD44^+^CD24^−^ breast CSCs, and induced the CSC differentiation into endothelial cells that express several endothelial markers (e.g., CD31, VE-Cadherin, CD105 and vWF). They also showed that breast CSCs gave rise to endothelial cells in NOD/SCID mice xenograft [55]. Although this paper was short in explaining the mechanisms of how BCSC differentiation into endothelial cells, they highlighted the further research direction for breast CSC derived endothelial cells.

### 3.2. CSCs Transdifferentiate into Pericytes

Anatomically, blood vessels in tumors consist of endothelial cells and pericytes [51,56]. The pericytes play a pivotal role in maintenance of the blood–brain barrier, facilitation of vessel maturation, and initiation of vessel sprouting. Pericytes communicate with endothelial cells to regulate endothelial homeostasis [51,54,56,57]. Absence or disfunction of pericytes is correlated with increased metastasis in colorectal, prostate, pancreatic, and breast cancers [58]. Cheng et al. demonstrated that GSCs have the ability to transdifferentiate into pericytes [59]. The authors employed a lineage-specific fluorescence reporter system to validate that GSCs have the capacity to transdifferentiate into pericytes in vivo. By analyzing clinical samples, they found that the majority of tumor pericytes carry the same genetic alternations as neoplastic cells. Elimination of GSCs derived pericytes inhibited tumor growth. The authors further found the GSCs are recruited towards endothelial cells via SDF-1-CXCR4 axis, then the TGF-β induces GSCs give rise to pericytes (Figure 2).

### 3.3. The Factors Affecting CSC Transdifferentiation

#### 3.3.1. Hypoxia

Hypoxia inducible factor-1 (HIF-1α), a key transcriptional regulator for gene expression in response to hypoxia [60,61,62,63], is essential for hematopoietic stem cell to maintain the cell cycle quiescence [64]. Moreover, it has been reported to regulate angiogenesis via releasing some cytokines such as VEGF, platelet-derived growth factor (PDGF), and angiopoietin-1 (Ang-1) [62,63,65]. Soda et al. [53] employed the GFP-reporter mouse model and revealed a new way of VEGF-independent but Hif-1 induced GSC transdifferentiate into ECs. This study suggested that hypoxia, partly through the activation of HIF-1α, played an important role in the differentiation of glioblastoma stem cells to endothelial cells (Figure 2).

#### 3.3.2. TGF-β

In addition to the essential roles in regulating differentiation, chemotaxis, proliferation, and activation of many immune cells, transforming growth factor-β (TGF-β) is also involved in maintaining cancer stemness and facilitating metastasis [66,67,68]. In Glioblastoma, TGF-β has been demonstrated to induce GSCs to transdifferentiate into pericytes [59]. They used Immunoblot to screen out the potential factors of inducing GSCs to transdifferentiate into pericytes, and found that the TGF-β greatly upregulates SAM expression in vitro. Furthermore, they also observed that coculture of GSCs and human brain microvascular endothelial cell line HBMEC enhanced the GSCs to transdifferentiate into pericytes and integrate into endothelial cells complexes, which were attenuated by the TGF-β antibody. In conclusion, these studies illustrated a clear route of GSC’s pericyte transdifferentiation: the endothelial cells recruit GSCs via the SDF-1/CXCR4 axis and then TGF-β induces GSC differentiation into pericytes. These studies strongly supported previous studies showing that TGF-β promotes breast cancer CSCs/early progenitor differentiation [69]. However, the source of TGF-β involved in GSC transdifferentiation was not demonstrated in these studies. Tumor cells could secrete TGF-β, and the immune cells that constitute the microenvironment could also secrete TGF-β. The identification of the source of TGF-β will provide more conveniences for the further research in the future (Figure 2).

#### 3.3.3. NOTCH1

The NOTCH1 signaling pathway is important for cell–cell communication as well as it plays a key role in the development and regulation of angiogenesis [70]. Hovinga et al. showed that the CD133^+^ glioblastoma stem cells participate in endothelial hyperplasia and structure vascular glomeruloid bodies in a special 3D explant model. Further, they found that antagonised Notch results in a decrease in self-renewal potential of tumor CD133^+^ cells as well as a decrease in the number of cells or downregulation of CD133 expression in GSCs [71]. Another group has further demonstrated that NOTCH regulates angiogenesis in details [43]. They employed the NOTCH pathway inhibitor to treat CD133^+^/CD144^−^ GSCs and observed that NOTCH inhibition resulted in significant suppression of the transition from CD133^+^/CD144^−^ GSCs to CD133^+^/CD144^+^ endothelial progenitors (Figure 2).

All these studies demonstrate that the NOTCH1 pathway also regulates the GSC differentiation into endothelial cells in glioblastoma.

## 4. Perspectives

In general, blood vessels play an important role in tumor development and metastasis, and the anti-vascular drugs such as Bevacizumab showed little effects in many cancer patients. The study of CSCs transdifferentiating to endothelial cells or pericytes can provide a new insight in the understanding of tumor progression and relapse.

A series of molecules such as hypoxia induced HIF-1α, TGF-β, and NOTCH1 have been reported to promote CSC transdifferentiate to endothelial cells or pericytes in glioblastoma, breast cancer, and renal carcinomas. In addition, the p38 through TGF-β and JNK signaling regulates mesenchymal stem/stromal cell transdifferentiate to endothelial cells in colon cancer [72]. Those studies focused on that tumor cells/cancer stem cells directly contribute to the construction of blood vessels via transdifferentiation into endothelial cells or pericytes, and enriched the understanding of angiogenesis in tumor microenvironment, and provided potential targets for clinical therapeutics and drug development.

It is noteworthy that the mechanisms of how CSCs transdifferentiated into endothelial cells or pericytes remain unclear in most of solid tumors except glioblastoma. Recently, a growing number of studies show that the tumor microenvironment is advantageous conditions for angiogenesis. Many immune cells which infiltrated into the tumor microenvironment secrete a variety of cytokines including TGF-β, SDF-1, PDGFB to promote the migration and transformation of cancer stem cells. Nevertheless, some researchers disagree with the concept of CSC transdifferentiate into ECs just because that the frequency of GSC-EC conversion was not defined, as well as, GSC derived ECs do not contain cancer genetic alterations in human GBMs [73,74]. Interestingly, Chao Sun et al. have already demonstrated that fusion of cancer stem cells and mesenchymal stem cells also contributes to glioma neovascularization in vivo [75].

In the last ten years, many researches have attempted to target cancer stem cells. Numerous studies indicate that some cytostatic drugs have shown the therapeutic potential for targeting cancer stem cells. Salinomycin, a commonly applied breast cancer chemotherapeutic drug, has shown strong activity to hinder the proliferation of various cancer cells, including those drug-resistant tumor cells and cancer stem cells (CSCs) [76,77]. Metformin, a cheap but effective chemical compound, used worldwide in the treatment of type 2 diabetes, was reported to restrain tumor progression significantly. Furthermore, a study [78] indicated that metformin significantly down-regulated the incidence of ovarian cancer in type 2 diabetic patients, and another study showed metformin decreased tumor progression via restraining cancer stem cells in vitro and in vivo [79]. As mentioned above, several proteins (e.g., TGF-β, NOTCH1) involved in CSC transdifferentiate into endothelial cells and pericytes could be used as therapeutic targets. It has been reported that in glioblastoma, blocking VEGF or VEGFR2 inhibits endothelium maturation but has no effect on the differentiation of CD133^+^ glioblastoma stem cells into endothelial progenitors. Moreover, they observed that NOTCH1 silence decreases the transition of CD133^+^ glioblastoma stem cells into endothelial progenitors [43]. Therefore, it is speculated that the combination of blocking cytokines with stem cell inhibitors can also decrease CSC derived endothelial cells or pericytes.

Although the transdifferentiation of CSCs into ECs or pericyte provides a new potential strategy in antitumor therapies, the detailed mechanisms of CSC transdifferentiation remain elusive. In the future, we should pay attention to the following the two aspects. First, CSC transdifferentiate into ECs or pericyte needs to be further studied in vivo using mouse models and clinical samples and lineage-tracing approaches in mice will greatly help us to understand when and where CSCs are transdifferentiated into ECs or pericytes. Second, it is of great significance to make a profound study of how we can specifically target the CSCs-associated vasculature to suppress the tumor progression.

In conclusion, the observations of CSCs participating in VM formation as well as the transdifferentiation of CSCs into ECs or pericytes in various tumors suggest that the multi-differentiation potentials of CSCs in tumors is far more complex than we expected. Although the mechanism of CSC transdifferentiation remain indistinct, it will advance our understanding of the plasticity of cancer stem cells, and may help with drug development and therapeutic strategies against tumors.

## Figures and Tables

**Figure 1 cells-10-01070-f001:**
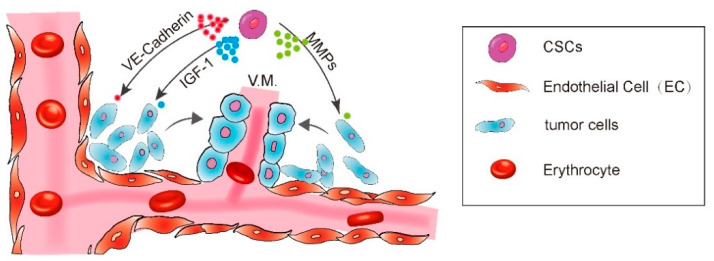
A diagram showing CSC enhance vasculogenic mimicry formation.

**Figure 2 cells-10-01070-f002:**
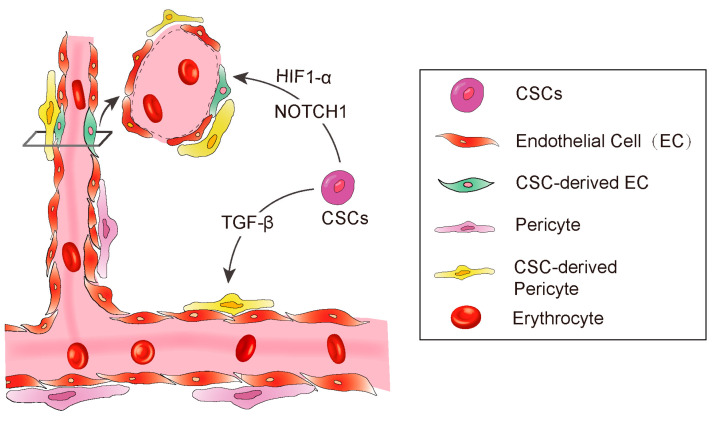
A diagram showing CSC derived angiogenesis. CSC-derived endothelial cells and pericytes play important roles in tumor vascular formation. Some signaling molecules, such as NOTCH1, HIF-1α, and TGF-β promote CSC to transdifferentiate into endothelial cells.

## Data Availability

Not applicable.

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
