# Peer review of "Cancer Stem Cells and Neovascularization"

_cells, 2021, doi:10.3390/cells10051070_

Round 1
Reviewer 1 Report
The manuscript, by an extensive literature review, is supporting the model of the VM for the spreading of the tumor bu CSCs.
The review is well written and easily read.
Author Response
Thank you very much for your encouragement and support.
Reviewer 2 Report
The review evaluates an interesting topic, the manuscript could however benefit from additional editorial work, preferably by a native English speaker. Several sentences are needlessly complicated, and includes irregular phrases (e.g. "recent knowledges", and "Bevacizumab would expect very limited effect").
Vascular mimicry is touted as key in tumor development, yet a discussion of why inhibition of VEGF by Bevacizumab failed to provide a favorable effect, is not included.
Next transdifferentiation of CSC into endothelial cells is highlighted with examples from different types of cancer. Only 1-2 references are given for each type of cancer, and the references deal with different markers. Consequently, no consensus is draw from comparing these studies, which appears as isolated stories.
In the Perspectives section is is noted that first, CSC to EC/pericyte needs to be studied further, then secondly, that we need to understand CSC to EC/pericyte transdifferentiation. A clear distinction between the two aspects (topics?) seems to be missing.
The figure is not well explained, and the legend need to be expanded to specify what the different colors of cells represent. Where are the CSCs located with respect to the blood vessel? If this is not an original figure, a reference needs to be included.
Author Response
Thank you very much for your precious comments.
point-by-point response:
1,Vascular mimicry is touted as key in tumor development, yet a discussion of why inhibition of VEGF by Bevacizumab failed to provide a favorable effect, is not included.
- We have added some new references to answer this question. The therapeutic effects of clinical anti-angiogenic agents on VM are very limited. Both the anti-VEGFA monoclonal antibody Bevacizumab and the endostatin have been re-ported to have no effect on VM.
2,Next transdifferentiation of CSC into endothelial cells is highlighted with examples from different types of cancer. Only 1-2 references are given for each type of cancer, and the references deal with different markers. Consequently, no consensus is draw from comparing these studies, which appears as isolated stories.
- Thank you for your comment. In this review, we would like to describe the phenomenon and associated mechanisms of CSCs directly take part in neovascularization. But there are limited previous papers available for references, and there are a small number of new research results in recent years, therefore, in our manuscript one kind of tumor has the isolate story.
3,In the Perspectives section is is noted that first, CSC to EC/pericyte needs to be studied further, then secondly, that we need to understand CSC to EC/pericyte transdifferentiation. A clear distinction between the two aspects (topics?) seems to be missing.
- Thank you for your comment. We have corrected the manuscript.
4,The figure is not well explained, and the legend need to be expanded to specify what the different colors of cells represent. Where are the CSCs located with respect to the blood vessel? If this is not an original figure, a reference needs to be included.
- Thank you for your kindly suggestions.
We drew new illustrations for this manuscript.
Reviewer 3 Report
The current review article entitled," Cancer stem cells and neovascularization” is quite interesting and has scientific merits to be considered for publication.
Authors summarize the roles of CSCs on tumor-associated angiogenesis via trans-differentiation or forming the capillary-like vasculogenic mimicry, as well as the roles of CSCs on facilitating endothelial cell-involved angiogenesis to support tumor progression and metastasis. Furthermore, they discuss the underlying regulation mechanisms, including the intrinsic signals of CSCs and the extrinsic signals such as cytokines from the tumor microenvironment.
Having said this, I dint find any novelty in the work since there are number of reviews available in literature. This can be improved if authors use schematic diagram to explain the under lying mechanism.
Below section should be explained more in mechanistic ways with recent references.
- CSCs transdifferentiate into endothelial cells
- The factors affecting CSC transdifferentiation should be elaborated more in depth with appropriated schematic diagram and more reference should be added..
Author Response
Thank you very much for your precious comments.
1, This can be improved if authors use schematic diagram to explain the under lying mechanism.
- Thank you for your kindly suggestions.
We drew new illustrations for this manuscript.
2, Below section should be explained more in mechanistic ways with recent references. CSCs transdifferentiate into endothelial cells...
The factors affecting CSC transdifferentiation should be elaborated more in depth with appropriated schematic diagram and more reference should be added..
- Thank you for your comment. In this review, we would like to describe the phenomenon and associated mechanisms of CSCs directly take part in neovascularization. Honestly, there are limited previous papers available for references, and there are a small number of new research results in recent years. in the revised manuscript ,we have added the new finding of cell fusion promotes angiogenesis in tumors.
Round 2
Reviewer 3 Report
By adding additional information Authors improved the manuscript. I have no further comment.